# Diverse and atypical manifestations of Q fever in a metropolitan city hospital: Emerging role of next-generation sequencing for laboratory diagnosis of *Coxiella burnetii*

**Fanfan Xing**[1], **Haiyan Ye**[1], **Chaowen Deng**[1], **Linlin Sun**[1], **Yanfei Yuan**[1], **Qianyun Lu**[1], **Jin Yang**[1], **Simon K. F. Lo**[1], **Ruiping Zhang**[2], **Jonathan H. K. Chen**[3], **Jasper F. W. Chan**[1,4], **Susanna K. P Lau**[4]*, **Patrick C. Y. Woo**[4]*

**1** Department of Clinical Microbiology and Infection Control, The University of Hong Kong—Shenzhen Hospital, Shenzhen, Guangdong, China, **2** Department of Pathology, The University of Hong Kong—Shenzhen Hospital, Shenzhen, Guangdong, China, **3** Department of Microbiology, Queen Mary Hospital, Hong Kong, China, **4** Department of Microbiology, Li Ka Shing Faculty of Medicine, The University of Hong Kong, Hong Kong, China

* skplau@hku.hk (SKPL); pcywoo@hku.hk (PCYW)

**Data Availability Statement:** All relevant data are within the manuscript and its Supporting Information files.

## Abstract

Although Q fever has been widely reported in the rural areas of China, there is a paucity of data on the epidemiology and clinical characteristics of this disease in large metropolitan cities. In this study, we profile the epidemiology and clinical manifestations of Q fever from a tertiary hospital in Shenzhen, a Southern Chinese metropolitan city with a large immigrant population from other parts of China. A total of 14 patients were confirmed to have Q fever during a nine-year-and-six-month period, five of whom were retrospectively diagnosed during case review or incidentally picked up because of another research project on unexplained fever without localizing features. Some patients had the typical exposure histories and clinical features, while a few other patients had rare manifestations of Q fever, including one with heart failure and diffuse intracapillary proliferative glomerulonephritis, a patient presenting with a spontaneous bacterial peritonitis-like syndrome, and another one with concomitant Q fever and brucellosis. Using a combination of clinical manifestation, inflammatory marker levels, echocardiographic findings and serological or molecular test results, nine, three and two patients were diagnosed to have acute, chronic and convalescent Q fever, respectively. Seven, five and two patients were diagnosed to have Q fever by serological test, nested real-time PCR and next-generation sequencing respectively. Diverse and atypical manifestations are associated with Q fever. The incidence of Q fever is likely to be underestimated. Next-generation sequencing is becoming an important diagnostic modality for culture-negative infections, particularly those that the physicians fail to recognize clinically, such as Q fever.

## Author summary

We describe the epidemiology and clinical manifestations of Q fever from a tertiary hospital in Shenzhen, a Southern Chinese metropolitan city in China. A total of 14 patients

**Funding:** This study was partly supported by Sanming Project of Medicine in Shenzhen, China (SZSM201911014, http://wjw.sz.gov.cn/ztzl/smgc/ ), and FX received the award for this study. The funders had no role in study design, data collection and analysis, decision to publish, or preparation of the manuscript.

**Competing interests:** The authors have declared that no competing interests exist.

were confirmed to have Q fever during this study period. Notably, five of them were retrospectively diagnosed during case review or incidentally picked up because of another research project on patients with unexplained fever. Interestingly, some patients had rare manifestations of Q fever, such as heart failure and diffuse intracapillary proliferative glomerulonephritis and spontaneous bacterial peritonitis. One patient had concomitant Q fever and brucellosis. Half of the patients were diagnosed by traditional serological test, while the other half by PCR or next-generation sequencing. Clinicians should have a high index of suspicion of Q fever because of its diverse and atypical manifestations. The incidence of Q fever is likely to be underestimated. Next-generation sequencing is becoming increasingly important for diagnosis of culture-negative infections.

## Introduction

Q fever is a zoonotic infection caused by a pleomorphic intracellular bacterium, *Coxiella burnetii*. Domestic animals, mainly sheep, goats and cattle, are the major source for human infection [1], with the bacterium present in the faeces, urine, milk and placenta of the infected animals. In addition, *C. burnetii* can also be found in many other wild and domestic animals such as horses, dogs, pigs, some birds, etc. [2]. The major route of transmission of *C. burnetii* to human is through inhalation of contaminated aerosols and dust particles, and less commonly by handling and ingestion of infected meat and milk. Therefore, those who are in close contact with the animals, such as farmers, abattoir workers and veterinarians are at highest risk. Clinical presentation of Q fever can be acute or chronic. The acute form of the disease usually presents as a self-limited non-specific febrile illness or atypical pneumonia, whereas the manifestation of the chronic form is more variable, including endocarditis, hepatitis, meningitis, encephalitis, osteomyelitis, etc. Notably, Q fever has become a notifiable disease in the United States since 1999 due to its potential as a biological warfare agent [3]. Traditionally, Q fever is diagnosed in the laboratory using serological test by detection of antibodies. Recently, molecular tests such as polymerase chain reaction (PCR) amplification of specific targets have also been employed for more rapid diagnosis of this condition [4].

Although Q fever has been widely reported in the rural areas of China [5], there is a paucity of data on the epidemiology and clinical characteristics of this disease in large metropolitan cities. Since it is relatively uncommon in modern cities, diagnosis is often difficult as most clinicians may be unaware of the diverse manifestations of the disease. Often, the disease may be treated without noticing the diagnosis through the prescription of empirical doxycycline for atypical pneumonia or fever without localizing features. In this study, we profile the epidemiology and clinical manifestations of Q fever from a tertiary hospital in Shenzhen, a Southern Chinese metropolitan city with a large immigrant population from other parts of China. In addition, the use of next-generation sequencing (NGS), the state-of-the-art and emerging technology in clinical microbiology, for laboratory diagnosis of Q fever as well as other culture-negative infectious disease syndromes is also discussed.

## Materials and methods

### Ethical statement

Ethics approval and exemption on patient consent for this retrospective study were endorsed by the Institutional Review Board of The University of Hong Kong—Shenzhen Hospital ([2021]161).

## Patients

This was a retrospective study conducted over a nine-year-and-six-month period (1 July 2012 to 31 December 2021) in The University of Hong Kong—Shenzhen Hospital. This 1,400-bed multi-specialty hospital was established in 2012 and provides primary to tertiary medical services to the residents of Shenzhen city in both inpatient and outpatient settings. Shenzhen is a Special Economic Zone with an estimated population of nearly 18 million people including a large migrant population from other regions in China. Geographically, it is located in the Guangdong Province, immediately north to Hong Kong. Affected by the policy of the government in mainland China, Shenzhen has been one of the fastest growing cities in the world during the 1990s. The clinical details, laboratory data and radiological findings of all patients with Q fever were retrieved from the hospital electronic record system and analysed. Clinical specimens, including the sera for indirect immunofluorescence assay and blood samples for nested real-time PCR and NGS analysis, were collected and handled according to standard protocols [6]. The diagnosis of acute, chronic and convalescent Q fever was made based on a combination of clinical presentation, inflammatory marker levels, echocardiographic findings and serological or molecular test results. Endocarditis was diagnosed using modified Duke's criteria [7].

## Indirect immunofluorescence assay

Q fever serology was performed in our laboratory since September 2020 using the indirect immunofluorescence assay (Focus Diagnostics, California, USA) for detection of human IgM antibodies to *C. burnetii* by a 2-stage "sandwich" principle, in which the wells of the slide was coated with *C. burnetii* phase I/II antigen and the presence of IgM detected with fluorescein-labeled antibody to IgM. The test was performed and results interpreted according to manufacturer's instructions. A serum titer of $\geq$1:16 to both phase I and phase II antigens strongly suggests recent *C. burnetii* infection, while that of <1:16 to both phase I and phase II antigens argues against recent *C. burnetii* infection. During acute infection, the IgM titers to phase II antigen are greater than those to phase I antigen; whereas during chronic infection or convalescent phase, the IgM titers to phase I antigen are greater than or equal to phase II antigen. Detection of IgG antibodies was not performed because of budget limitations.

## Nested real-time PCR

Nested real-time PCR for *C. burnetii* was performed in our laboratory since August 2021 by targeting the transposon-like repetitive region, *IS1111* gene, according to a published protocol, with modifications [8]. Briefly, total nucleic acid was extracted from 300 μL of plasma using the MagaBio plus Virus DNA/RNA Purification Kit III (BIOER, Hangzhou, China). The nucleic acid was eluted in 60 μL of RNase-free water and was used as the template for nested real-time PCR. The primers and probe sequences of the nested real-time PCR assay were synthesized by BGI (Beijing, China) (S1 Table). Real-time PCR was performed using the Quanti-Nova Probe PCR Kit (Qiagen) and in a QuantStudio 5 Real-Time PCR Instrument (ABI, Singapore). The master mix and cycling conditions are shown in S2 and S3 Tables.

## Next-generation sequencing

Ethylene Diamine Tetraacetic Acid (EDTA)-treated blood was collected from the patients and sent to the BGI PathoGenesis Pharmaceutical Technology Co., Ltd (Shenzhen, China) for NGS analysis of pathogenic microorganisms.

## Results

### Clinical characteristics

A total of 14 patients were confirmed to have Q fever during the study period (Table 1). Twelve patients were males and two were females. The median age was 46.5 (range 20–65). Three had high risk occupations (chef in case 6 and farmers in cases 10 and 11). Four (cases 2, 6, 10 and 11) had clear histories of recent exposure to goat, sheep or cattle and 4 others (cases 3, 8, 9 and 12) have recent visit to the rural environment. The remaining 6 patients (cases 1, 4, 5, 7, 13 and 14) denied any recent contact with livestock, although case 1 had recent unprotected sexual intercourse, which has been reported to be a possible route of *C. burnetii* transmission [9]. The median interval between disease onset and hospital admission was 10 (range 6–90) days and that between hospital admission and confirmation of the diagnosis of Q fever was 10.5 (range 3–600) days. All the 14 patients presented with fever and non-specific symptoms, although cases 2 and 3 had very severe headache and were admitted to the neurology unit as suspected meningitis. Case 6 presented with symptoms of heart failure and glomerulonephritis (Fig 1) and case 9 presented with a spontaneous bacterial peritonitis-like syndrome (Fig 2). Four (cases 1, 2, 6 and 9) and 9 (cases 1, 2, 3, 4, 6, 7, 8, 10 and 14) patients had hepatomegaly and splenomegaly, respectively. Using a combination of clinical manifestation, inflammatory marker levels, echocardiographic findings and serological or molecular test results, 9 (cases 1, 2, 3, 4, 5, 7, 8, 11 and 12), 3 (cases 6, 9 and 14) and 2 (cases 10 and 13) patients were diagnosed to have acute, chronic and convalescent Q fever, respectively. All the 14 patients survived. For the 10 patients (cases 1, 2, 3, 4, 5, 6, 8, 9, 12 and 13) who had fever on admission, the median time to defervescence was 3.5 (range 1–7) days.

### Laboratory findings

The laboratory findings of the 14 patients with Q fever in the present cohort are summarized in Table 2. Three of the 10 patients (cases 1, 9 and 12) had increased peripheral white cell count and neutrophilia. Five patients (cases 2, 3, 5, 6 and 10) had moderate thrombocytopenia. Twelve (cases 1, 2, 3, 4, 5, 7, 8, 9, 10, 12, 13 and 14) had mildly to moderately elevated liver parenchymal enzymes. The median (range) serum alanine transaminase and aspartate transaminase levels were 103.2 (11.1–154) U/L and 54.9 (25–167.9) U/L respectively. The median erythrocyte sedimentation rate (ESR) was 28 (range 5–111) mm/hour, with 6 patients (cases 1, 4, 5, 6, 7 and 11) having moderately raised ESR and one patient (case 12) with an ESR of >100 mm/hour. The median C-reactive protein (CRP) was 87.5 (range 0.45–219.2) mg/L with 13 patients having elevated CRP. The median activated partial thromboplastin time (aPTT) was 46.1 (range 33.9–82.3) seconds, with 10 patients (cases 1, 4, 5, 6, 7, 8, 9, 12, 13 and 14) having prolonged aPTT. Lupus anticoagulant was checked in 9 patients and 6 (cases 1, 6, 7, 9, 12, and 14) were detected.

### Echocardiography findings and endocarditis

Echocardiography was performed in all of the 14 patients, with 6 of them showing abnormal findings (Table 1). According to the modified Duke's criteria, 2 patients (cases 6 and 14) fulfilled the criteria for infective endocarditis.

### Microbiological findings and laboratory diagnosis of Q fever

Seven patients (cases 1, 6, 9, 10, 11, 12 and 13) were diagnosed to have Q fever by positive serological test (Table 2). Five patients (cases 4, 5, 7, 8 and 14) were diagnosed by positive nested

**Table 1. Demographic and clinical characteristics of patients in the present cohort.**

| Patient No. | Year of diagnosis | Sex/Age | Occupation | Exposure history | Interval between disease onset to hospital visit (days) | Interval between hospitalization and diagnosis (days) | Form of Q fever | Underlying disease | Clinical manifestation | Chest radiographic finding | Abdominal imaging finding | Echocardiography | Days from antibiotic treatment to defervescence |
|---|---|---|---|---|---|---|---|---|---|---|---|---|---|
| 1 | 2014 | M/27 | Docker | Unprotected sexual exposure | 11 | 8 | Acute | None | Fever, chills, weakness, arthralgia, myalgia, relative bradycardia, hepatomegaly, splenomegaly | None | Gallbladder wall thickening, hepatomegaly, splenomegaly | Normal | 6 |
| 2 | 2019 | M/37 | Engineer | Dog, goat meat, rural environment | 6 | 3 | Acute | Hypertension | Fever, chills, night sweats, weakness, headache, arthralgia, myalgia, nausea, vomiting, abdominal pain, lower back pain, cough, conjunctival congestion, relative bradycardia, jaundice, hepatomegaly, splenomegaly | None | Hepatomegaly, splenomegaly, kidney stone | Normal | 2 |
| 3 | 2019 | M/65 | Headmaster | Guinea pigs, hens, rural environment | 7 | 5 | Acute | Hypertension, secondary hypothyroidism | Fever, weakness, headache, arthralgia, myalgia, conjunctival congestion, relative bradycardia, splenomegaly | Bilateral patchy infiltrates and atelectasis | Splenomegaly | Sclerosis of aortic valves | 3 |
| 4 | 2020 | M/20 | Student | Unclean food | 9 | 600 | Acute | None | Fever, chills and rigors, splenomegaly | Normal | Splenomegaly | Normal | 1 |
| 5 | 2020 | M/40 | Unemployed | Dogs, rabbits | 6 | 570 | Acute | None | Fever, skin rash, chills, general pain | Multiple pulmonary bullae | No abnormality | Normal | 3 |
| 6 | 2020 | M/62 | Chef | Livestock, rural environment | 7 | 12 | Chronic | Hypertension, congestive heart failure | Fever, facial puffiness, lower limb edema, night sweats, weakness, abdominal pain, cough, dyspnea, relative bradycardia, lymphadenopathy, hepatomegaly, splenomegaly | Bilateral patchy infiltrates and pleural effusion | Splenomegaly, enlarged bilateral kidneys | Thickening of mitral and tricuspid valves and chordae tendineae; aortic valves stenosis with insufficiency and suspected abscess or hematoma; pericardial effusion | 4 |
| 7 | 2020 | M/35 | Clerk | None | 14 | 510 | Acute | None | Fever, dizziness | Inflammation in bilateral lower lung and the left lingual lobe | Cholecystitis, ascites, left kidney stone, splenomegaly | Normal | Still afebrile when discharged |

*(Continued)*

Table 1. (Continued)

| Patient No. | Year of diagnosis | Sex/Age | Occupation | Exposure history | Interval between disease onset to hospital visit (days) | Interval between hospitalization and diagnosis (days) | Form of Q fever | Underlying disease | Clinical manifestation | Chest radiographic finding | Abdominal imaging finding | Echocardiography | Days from antibiotic treatment to defervescence |
|---|---|---|---|---|---|---|---|---|---|---|---|---|---|
| 8 | 2021 | M/44 | Clerk | Lizards, tortoise, fresh water fish, crickets, bovine placenta; rural environment | 6 | 300 | Acute | None | Fever, headache, nausea and vomiting | Micronodules seen in the left lung, lymph nodes or inflammatory granulomas suspected | Splenomegaly | Normal | 2 |
| 9 | 2021 | M/35 | Company manager | Rural environment | 10 | 6 | Chronic | Fatty liver | Fever, chills, weakness, abdominal pain, relative bradycardia, hepatomegaly | Bilateral pleural effusion and atelectasis | Gallbladder wall thickening, hepatomegaly, fatty liver, thickened capsule of bilateral kidney, peritonitis | Normal | 7 |
| 10 | 2021 | F/49 | Farmer | Goat | 20 | 9 | Convalescent | Hypertension | Fever[a], night sweats, arthralgia, myalgia, splenomegaly | None | Liver cyst, splenomegaly | Enlargement of left atrium | - |
| 11 | 2021 | M/50 | Farmer | Goat | 90 | 30 | Acute | None | Fever[a], night sweats, arthralgia, low back pain | None | Inflammation of terminal ileum | Diastolic dysfunction of left ventricle | - |
| 12 | 2021 | M/52 | Government servant | Cat, rural environment | 21 | 8 | Acute | Hypertension, diabetes mellitus, gout | Fever, weakness, rash, chest pain, lymphadenopathy | Bilateral consolidation and pleural effusion | Thickened capsule of bilateral kidneys | Regurgitation of mitral and tricuspid valves; pericardial effusion | 4 |
| 13 | 2021 | M/56 | Unemployed | None | 51 | 45 | Convalescent | Chronic obstructive pulmonary disease | Fever, weakness, low back pain, relative bradycardia | None | None | Normal | 5 |
| 14 | 2021 | F/56 | Retired clerk | Dog | 10 | 4 | Chronic | Hypertension | Fever[a], chills, headache, splenomegaly | None | Cholecystectomy, splenomegaly | Vegetation of aortic valves; pericardial effusion | - |

[a]These patients became afebrile before admission and commencement of antibiotic.

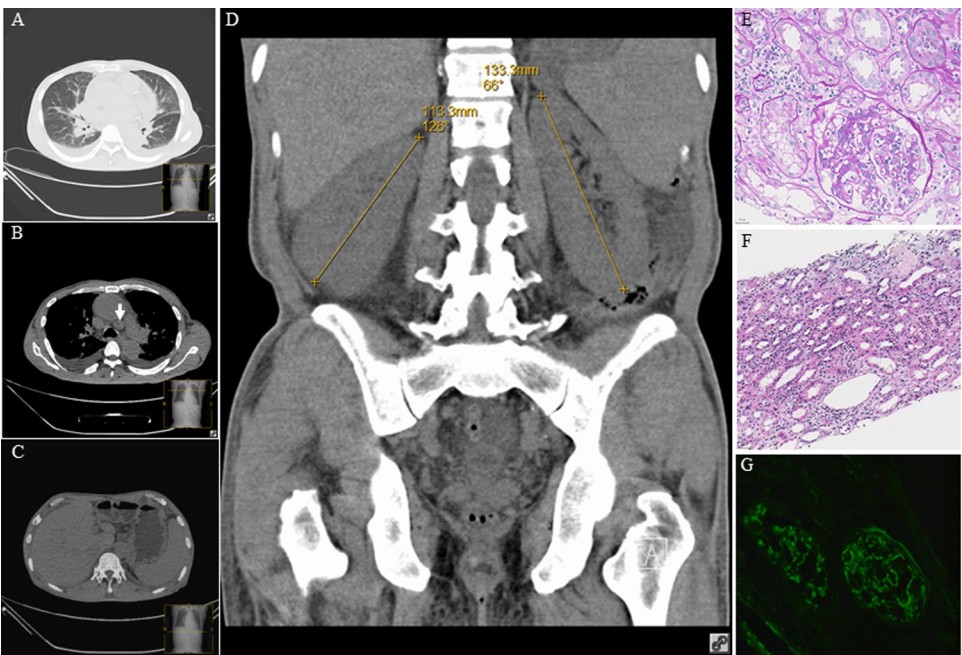

**Fig 1. Computed tomography of the thorax and abdomen and histology of renal biopsy for Case 6.** (A) Bilateral diffuse interstitial infiltrates pleural effusion. (B) Bilateral pleural effusion and mediastinal lymphadenopathy (arrow). (C) Hepatosplenomegaly and ascites. (D) Symmetrically enlarged kidneys. (E) Diffuse intracapillary hyperplasia in the glomerulus with neutrophil infiltration in the capillary lumen, and mild proliferation of mesangial cells and stroma in focal segments of the glomerulus (PAS×400). (F) Focal renal interstitial fibrosis and edema with neutrophil, lymphocyte and plasmacyte infiltration (H&E×200). (G) Granular C3 deposition in the capillary wall and mesangial regions on immunofluorescent staining (×200).

real-time PCR and two (cases 2 and 3) were diagnosed by NGS. Ten patients had brucella serology performed and was positive in one (case 13).

## Discussion

In this study, we describe the diverse and some atypical manifestations of Q fever in a densely populated metropolitan city. In the present cohort, some patients had the typical occupation,

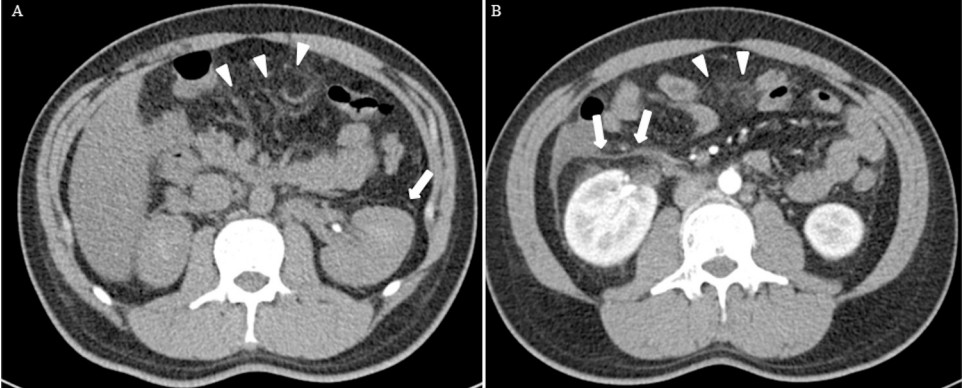

**Fig 2. Computed tomography of the abdomen for Case 9.** (A) Plain film showing peritonitis (arrowhead) and thickened capsule of the left kidney (arrow). (B) Contrast-enhanced image (arterial phase) showing peritonitis (arrowhead) and thickened capsule of the right kidney (arrow).

**Table 2. Laboratory findings of patients in the present cohort.**

| Patient No. | WBC (×10⁹/L) | Neutrophil (×10⁹/L) | Platelet (×10⁹/L) | ALT (U/L) | AST (U/L) | Tbil (mmol/L) | ESR (mm/h) | CRP (mg/L) | PT (s) | aPTT (s) | Lupus anticoagulant | Anti-cardiolipin IgM (U/mL) | Anti-cardiolipin IgG (U/mL) | Anti-MPO-IgG (RU/mL) | Anti-PR3-IgG (RU/mL) | RF (U/mL) | Brucella Ab | Diagnostic test for Q fever |
|---|---|---|---|---|---|---|---|---|---|---|---|---|---|---|---|---|---|---|
| 1 | 10.9 | 7.9 | 366 | 154 | 42 | 15.4 | 57 | 86 | 14.2 | 54 | Detected | 213 | 84.4 | 26.1 | 46.2 | 17 | Negative | CF: phase II 1:640 IFA IgM: phase II 1:800 |
| 2 | 4.6 | 3 | 68 | 107.5 | 100.7 | 53.1 | 17 | 179.5 | 14.4 | 39.4 | Not done | Not done | Not done | Not done | Not done | Not done | Negative | NGS 211 sequences detected |
| 3 | 5.5 | 4.2 | 114 | 46.6 | 30.9 | 15.2 | 17 | 54 | 16.7 | 36.1 | Not done | <2 | <2 | 2.05 | <2 | Not done | Negative | NGS 1021 sequences detected |
| 4 | 5.38 | 4.16 | 196 | 110.6 | 51.9 | 14 | 29 | 68.61 | 14 | 45 | Not done | Not done | Not done | 5.45 | 3.01 | Not done | Not done | Nested real-time PCR positive |
| 5 | 3.75 | 2.87 | 103 | 124.6 | 139 | 10.7 | 40 | 98.57 | 13.7 | 46.3 | Not done | Not done | Not done | Not done | Not done | 8.6 | Not done | Nested real-time PCR positive |
| 6 | 7 | 4.6 | 94 | 11.1 | 25 | 12.7 | 33 | 32 | 13.7 | 49.8 | Detected | 7.89 | 3.67 | <2 | 4.71 | 43.9 | Not done | IFA IgM: phase I > 1:8192, phase II 1:512 |
| 7 | 8.54 | 5.12 | 178 | 86.7 | 56.4 | 26.4 | 51 | 122.91 | 16.8 | 51.9 | Detected | > 800 | > 800 | 76.3 | > 800 | 21.1 | Negative | Nested real-time PCR positive |
| 8 | 6.64 | 4.91 | 220 | 138.4 | 95 | 30.9 | 20 | 89.05 | 14.4 | 45.9 | Not detected | Negative | Negative | Not done | Not done | Not done | Negative | Nested real-time PCR positive |
| 9 | 19 | 15.3 | 290 | 152.5 | 88 | 16.5 | 22 | 124.9 | 14.9 | 60.9 | Detected | 88.2 | >480 | 10.9 | 25.8 | 18.9 | Not done | IFA IgM: phase I 1:1024, phase II 1:1024 |
| 10 | 3.5 | 2.3 | 137 | 76.7 | 53.4 | 7.4 | 5 | 0.45 | 12.4 | 33.9 | Not detected | Negative | Negative | Not done | Not done | 48.1 | Negative | IFA IgM: phase I 1:128, phase II 1:64 |
| 11 | 5.72 | 4.55 | 158 | 33 | 29.5 | 5 | 28 | 12.1 | 13.6 | 36.6 | Not detected | Negative | Negative | Not done | Not done | 8.8 | Negative | IFA IgM: phase I negative, phase II 1:64 |
| 12 | 13.1 | 11.2 | 281 | 98.8 | 75.9 | 26.9 | 111 | 219.2 | 15.5 | 82.3 | Detected | 11.3 | 99.4 | 11.8 | 35.7 | 12.1 | Negative | IFA IgM: phase I 1:2048, phase II 1:8192 |
| 13 | 6.5 | 3.4 | 410 | 52.4 | 42.1 | 5.2 | 7 | 40.7 | 12.5 | 41.1 | Not done | <2 | <2 | <2 | 2.07 | N/A | 1:200 | IFA IgM: phase I 1:128; phase II 1:16 |

(*Continued*)

**Table 2.** (Continued)

| Patient No. | WBC (×10⁹/L) | Neutrophil (×10⁹/L) | Platelet (×10⁹/L) | ALT (U/L) | AST (U/L) | Tbil (mmol/L) | ESR (mm/h) | CRP (mg/L) | PT (s) | aPTT (s) | Lupus anticoagulant | Anti-cardiolipin IgM (U/mL) | Anti-cardiolipin IgG (U/mL) | Anti-MPO-IgG (RU/mL) | Anti-PR3-IgG (RU/mL) | RF (U/mL) | Brucella Ab | Diagnostic test for Q fever |
|---|---|---|---|---|---|---|---|---|---|---|---|---|---|---|---|---|---|---|
| 14 | 3.3 | 2.1 | 151 | 144.8 | 167.9 | 5.3 | Not done | 143.6 | 12.5 | 55.9 | Detected | 293.2 | 34.8 | 4.98 | 16.6 | 10.5 | Negative | IFA IgM: phase I and II not detected, nested real-time PCR positive |

Abbreviation: PCR: polymerase chain reaction; NGS: next-generation sequencing; IFA: immunofluorescence assay; ALT: alanine transaminase; AST: aspartate transaminase; Tbil: total bilirubin; ESR: erythrocyte sedimentation rate; CRP: C-reactive protein; PT: prothrombin time; aPTT: activated partial thromboplastin time; anti-MPO-IgG: anti-myeloperoxidase-IgG; anti-PR3-IgG: anti-proteinase 3-IgG; RF, rheumatoid factor.

exposure history and manifestation. For example, cases 10 and 11 were a couple and they were farmers with clear contact history with goats. At the same time, a few other patients have atypical and rare manifestations of Q fever. Case 9 was a 35-year-old man with underlying fatty liver who presented with fever, chills and abdominal pain. Although the clinical diagnosis was spontaneous bacterial peritonitis, chest radiograph revealed bilateral pleural effusion and atelectasis and contrast computed tomography (CT) of the abdomen showed abdominal effusion, thickening of parietal peritoneum and bilateral renal capsules (Fig 2). In addition, he had prolonged aPTT at 60.9 seconds and he failed to respond to empirical intravenous piperacillin-tazobactam for the treatment of spontaneous bacterial peritonitis. Although trans-esophageal echocardiography did not show any vegetation, Q fever serology was performed and revealed high titers (both ≥1:1024) of IgM to both phase I and phase II antigens. In the literature, only one other case of Q fever with a spontaneous bacterial peritonitis-like syndrome was reported [10]. In that 55-year-old man with underlying type 2 diabetes mellitus, he presented with fever and chills for 20 days but there was no abdominal pain. Only diffuse abdominal fullness without tenderness was observed during physical examination. Similar to our patient, he also had prolonged aPTT of 74.5 seconds and mildly deranged liver function test. CT of the abdomen did not show any ascites but gallium scan revealed hepatomegaly with diffuse uptake in the abdomen, suggestive of peritonitis or peritoneum carcinomatosis. Q fever serology subsequently showed high titers (both ≥1:2560) of IgG and IgM to phase II antigens. In addition to this case 9 of Q fever presenting as spontaneous bacterial peritonitis, the manifestation of case 6 was also uncommon. Case 6 was a 62-year-old man with underlying hypertension and congestive heart failure who presented with facial puffiness and bilateral lower limb swelling for 5 days without fever. Serum creatinine was elevated and on increasing trend and there was hypoalbuminemia and microscopic hematuria. CT of the thorax and abdomen showed interstitial pulmonary edema, pericardial and bilateral pleural effusion, mediastinal lymphadenopathy, bilateral enlarged kidneys and ascites (Fig 1A–1D). Q fever serology showed that the titers of IgM to phase I and phase II antigens were 1:8192 and 1:512 respectively. Histological examination of the renal biopsy revealed diffuse intracapillary proliferative glomerulonephritis (Fig 1E–1G), which has been reported only once as a complication of Q fever [11]. Other reported cases of glomerulonephritis associated with Q fever were mainly focal and segmental proliferative glomerulonephritis, mesangioproliferative glomerulonephritis, mesangiocapillary glomerulonephritis and membranoproliferative glomerulonephritis [11–14]. In our patient, the glomerulonephritis and renal function responded promptly to doxycycline treatment of the Q fever.

The incidence of Q fever is underestimated. Failure to make a diagnosis of Q fever is mainly due to the difficulty for the clinician to recognize the disease or lack of laboratory support to confirm the diagnosis. In modern cities where farms are not commonly found and the incidence of Q fever low, doctors are unfamiliar with the diverse presentations of this infection. Moreover, the disease is often self-limited or if presented as atypical pneumonia, it may be treated empirically with doxycycline without confirming the microbiological diagnosis through ordering the appropriate laboratory tests. As illustrated in the present cohort, case 13 was a 56-year-old man presented with fever and back pain. As *Brucella melitensis* was isolated from the patient's blood culture and brucella serology was also positive, the patient was treated with doxycycline and gentamicin for one week followed by doxycycline for five more weeks. The patient responded and was discharged uneventfully. It was only during case review one and a half months later that the diagnosis of Q fever was also suspected. The serum of the patient was retrieved and Q fever serology showed that the titers of IgM to phase I and phase II antigens were 1:128 and 1:16 respectively. In fact, co-infection of *C. burnetii* and *Brucella* species has only been reported once in the literature [15]. In that case, the patient was a 30-year-

old agricultural worker who presented with fever and non-specific symptoms. He worked in a sheep farm and has consumed unpasteurized dairy products of sheep origin in Bosnia and Herzegovina. Similar to our case 13, blood culture was positive for *B. melitensis* and brucella serology was also positive. In addition, *C. burnetii* phase II IgM/IgG titers were 1:50 and 1:1024, respectively, confirming the co-infection. As the animal source of these two bacteria are common, we speculate that *C. burnetii* and *Brucella* co-infection is also under reported, as patients who are treated with brucellosis would have their Q fever treated automatically. In addition to case 13, it is of note that four other patients (cases 4, 5, 7 and 8) were clinically diagnosed to have typhus-like illness during their admissions, although none of them was laboratory confirmed. Hence, doxycycline was empirically prescribed and they responded promptly. Their diagnosis of Q fever was only incidentally confirmed by real-time quantitative PCR when they were investigated retrospectively for unexplained fever without localizing features in another research project. As for the lack for laboratory support, some microbiology laboratories are not equipped with tests for Q fever. For example, for the laboratory in our hospital, serology test was only available since late 2020. This is indeed the reason why 70% of the Q fever cases in the present cohort were made since this time. For case 1 which the diagnosis was made in 2014, the laboratory test was actually carried out in Hong Kong when the diagnosis of Q fever was suspected despite there was no obvious exposure histories to animals.

NGS is becoming an important diagnostic modality for culture-negative infections, particularly those that the physicians fail to recognize clinically. When NGS technologies first appeared in the market, they were mainly used for genome sequencing. With the advancement of sequencing chemistries and computational capacity, NGS technologies have matured into clinical applications in the recent years [16]. In the clinical setting for infectious diseases, NGS is used most often for patients who have fever without localizing features or culture-negative infections. We have recently reported its application in fungal diagnosis as well as confirming the first case of listeria meningitis in a patient with autoantibody against interferon gamma and another one with *Mycobacterium marinum* infection [17–19]. In the present cohort, case 2 and 3 both presented with fever and severe headache and were admitted to the neurology unit as suspected meningitis. Lumbar puncture was performed but analysis of the cerebrospinal fluid was negative. At that time, Q fever serology and real-time PCR test were not yet available in our hospital. Hence, blood samples of the patients were sent for NGS, which revealed 211 and 1021 sequence reads of *C. burnetii* respectively, confirming the diagnosis of Q fever. In our setting, the NGS was performed in a private laboratory with the cost of RMB 4,500 (~698 USD) per sample and the turn-around-time for these two cases were two days, making the use of this robust technology pragmatic and affordable in the clinical setting. It is of note that Q fever has been diagnosed a few times using NGS in the literature [20–22], including a recent outbreak in southern China [21]. In that outbreak, plasma samples from 138 out of 2382 patients who had fever of unknown source were tested positive for *C. burnetii* sequences by NGS and the outbreak was finally traced to goats and cattle in a slaughterhouse [21]. With its low equipment cost, short turn-around-time and portable size, the recent invention of the Oxford Nanopore Technologies' MinION device and further improvement of its sequencing accuracy will make the use of NGS within clinical microbiology laboratories feasible in the near future.

## Supporting information

**S1 Table. Primers and probe for *Coxiella burnetii IS*1111 gene nested real-time PCR.** (DOCX)

**S2 Table. Master mix for *Coxiella burnetii IS*1111 gene nested real-time PCR.**
(DOCX)

**S3 Table. Cycling profile of *Coxiella burnetii IS*1111 gene nested real-time PCR.**
(DOCX)

## Acknowledgments

We are grateful to the staff at the Department of Clinical Microbiology and Infection Control, the Department of Pathology, the Department of Medical Imaging, The University of Hong Kong-Shenzhen Hospital, and the Department of Microbiology, The University of Hong Kong for facilitation of the study.

## Author Contributions

**Conceptualization:** Fanfan Xing, Jasper F. W. Chan, Susanna K. P Lau, Patrick C. Y. Woo.

**Data curation:** Fanfan Xing, Haiyan Ye, Chaowen Deng, Linlin Sun, Jin Yang, Patrick C. Y. Woo.

**Formal analysis:** Fanfan Xing, Haiyan Ye, Jonathan H. K. Chen, Patrick C. Y. Woo.

**Funding acquisition:** Fanfan Xing.

**Investigation:** Fanfan Xing, Chaowen Deng, Linlin Sun, Yanfei Yuan, Qianyun Lu, Jin Yang, Simon K. F. Lo, Ruiping Zhang, Jasper F. W. Chan.

**Methodology:** Fanfan Xing, Yanfei Yuan, Qianyun Lu, Simon K. F. Lo, Ruiping Zhang, Jonathan H. K. Chen, Jasper F. W. Chan, Susanna K. P Lau, Patrick C. Y. Woo.

**Project administration:** Simon K. F. Lo, Jasper F. W. Chan.

**Resources:** Simon K. F. Lo.

**Supervision:** Jasper F. W. Chan, Susanna K. P Lau, Patrick C. Y. Woo.

**Writing – original draft:** Fanfan Xing, Jonathan H. K. Chen, Susanna K. P Lau, Patrick C. Y. Woo.

**Writing – review & editing:** Fanfan Xing, Haiyan Ye, Chaowen Deng, Linlin Sun, Yanfei Yuan, Qianyun Lu, Jin Yang, Simon K. F. Lo, Ruiping Zhang, Jonathan H. K. Chen, Jasper F. W. Chan, Susanna K. P Lau, Patrick C. Y. Woo.

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
