## [Decision Letter · Decision Letter 0]

10 Mar 2022

Dear Professor Woo,

Thank you very much for submitting your manuscript "Diverse and atypical manifestations of Q fever in a metropolitan city hospital: emerging role of next-generation sequencing for laboratory diagnosis of Coxiella burnetii, a potential biological warfare agent" for consideration at PLOS Neglected Tropical Diseases. As with all papers reviewed by the journal, your manuscript was reviewed by members of the editorial board and by several independent reviewers. The reviewers appreciated the attention to an important topic. Based on the reviews, we are likely to accept this manuscript for publication, providing that you modify the manuscript according to the review recommendations. 

Sincerely,

Joseph M. Vinetz

Deputy Editor

Joseph Vinetz

Deputy Editor

Reviewer's Responses to Questions

**Key Review Criteria Required for Acceptance?**

**Methods**

-Are the objectives of the study clearly articulated with a clear testable hypothesis stated?

-Is the study design appropriate to address the stated objectives?

-Is the population clearly described and appropriate for the hypothesis being tested?

-Is the sample size sufficient to ensure adequate power to address the hypothesis being tested?

-Were correct statistical analysis used to support conclusions?

-Are there concerns about ethical or regulatory requirements being met?

Reviewer #1: (No Response)

Reviewer #2: Excellent

**Results**

-Does the analysis presented match the analysis plan?

-Are the results clearly and completely presented?

-Are the figures (Tables, Images) of sufficient quality for clarity?

Reviewer #1: (No Response)

Reviewer #2: Excellent

**Conclusions**

-Are the conclusions supported by the data presented?

-Are the limitations of analysis clearly described?

-Do the authors discuss how these data can be helpful to advance our understanding of the topic under study?

-Is public health relevance addressed?

Reviewer #1: (No Response)

Reviewer #2: Excellent

**Editorial and Data Presentation Modifications?**

Reviewer #1: (No Response)

Reviewer #2: Minor revision

**Summary and General Comments**

Reviewer #1: In this work, the authors describe the symptoms of Q-fever in 14 cases from a southern Chinese city. This work contributes to our understanding of Q-fever symptoms. The title and some discussion in the text seem to suggest that next gen sequencing played a major role in this study and evidence will be presented about its utility when traditional methods fail. This however is not the case and I would suggest removal of this aspect from the title and some of the discussion. I am not advocating that the authors remove the use of next gen sequencing in their methods – it just appears to be over-hyped in the discussion. There are various important aspects of the methods (biological samples used for DNA analyses, and sequencing details) that are missing and thus make it difficult to evaluate this work. Overall, the report is well written, but certain minor English problems are prevalent. Also, in some cases, sentences contain a mixture of seemingly unrelated concepts that make it difficult to precisely identify the point being made by the authors. 

Q fever is thought to be underdiagnosed throughout the world due to the diversity of non-specific symptoms. What is different about these cases compared to previously reported symptoms. Authors mention “atypical” manifestations, but the reader has to get to the discussion (line 225) to find the first comparison to existing literature.

Authors discuss the patients and source of their data under “Patients” (Materials and Methods), and this seems to only include electronic records. Where did the material used for serological and DNA-based testing come from? Perhaps some of the results were in electronic records, but what about the material that was processed in the author’s laboratory?

Why did the authors use IgM instead of IgG as IgM is more less specific and prone to false positives?

The IS1111 region is a great target for detection and multiple assays have been designed. Authors do not cite what assay they are using. This is important as previous work should have been performed to validate the assay and determine sensitivity/specificity.

The use of next-gen sequencing needs elaboration. What sequencing platform was used, what kit was used, what sequencing depth was obtained? Bioinformatic methods are also completely missing - methods for filtering human sequencing, methods for identifying pathogen sequences, methods of ensuring that putative pathogen sequences are specific to Cb. Results on next gen sequencing are also missing – what sequences were obtained, what were the quality of the reads, what depth and breadth of Cb coverage was obtained? Including simply the number of reads that can be attributed to Cb is just not enough to convince anyone that they are from Cb – especially if the highly sensitive IS1111 pcr was negative. How do we know that these reads are not from bacterial amplification clones, adaptors, or other artificial sequence?

Was serology, pcr, and sequencing performed on each sample (are only positive results shown)? 

The figures are not referenced in results section of the text (only in the discussion).

Specific comments:

Title: the fact that Cb is a potential biological weapon has no bearing on this work and should be removed from the title.

Abstract: 

Line 34: Rather than “typical occupation and exposure history”, do the authors mean “typical symptoms”?

Line 35: Should be in the past tense – change “have” to “had”.

Line 41: Is this for the initial diagnosis? Was PCR and sequencing done on all patients? 

Line 42: This sentence needs to be clarified. I don’t think that the authors are contending that the disease was self-limiting because of atypical manifestations (as the sentence currently reads). There are a lot of concepts tied into this sentence – there are aspects of Q that cause incidence rates to be underestimated – and there is the treatment and duration of the disease. Can these concepts be separated?

Line 46 (and line 60 and 90): Why is “culture-negative infections” included here as Cb is extremely difficult and time consuming to culture and probably not attempted in clinics for routine diagnosis? Do authors intend to refer to bacteria that cannot be cultured?

Line 58: What is a high “index of suspicion”?

Line 70: Ingestion is not a “major route of transmission” for Q.

Line 122: Change “great” to “greater”.

Line 145 (and throughout): Make “case” plural when appropriate.

Line 213 and 214: I don’t understand why authors contrast typical exposure (to animals) with atypical manifestations.

Reviewer #2: The manuscript is excellent and should be published

PLOS authors have the option to publish the peer review history of their article (what does this mean?). If published, this will include your full peer review and any attached files.

Reviewer #1: No

Reviewer #2: Yes: Dr. Ulrich Wernery

Figure Files:

Data Requirements:

Reproducibility:

References

---

## [Editor Report · Decision Letter 1]

28 Mar 2022

Dear Prof. Woo,

We are pleased to inform you that your manuscript 'Diverse and atypical manifestations of Q fever in a metropolitan city hospital: emerging role of next-generation sequencing for laboratory diagnosis of Coxiella burnetii' has been provisionally accepted for publication in PLOS Neglected Tropical Diseases.

Best regards,

Joseph M. Vinetz

Deputy Editor

Joseph Vinetz

Deputy Editor

---

## [Editor Report · Acceptance letter]

14 Apr 2022

Dear Prof. Woo,

We are delighted to inform you that your manuscript, "Diverse and atypical manifestations of Q fever in a metropolitan city hospital: emerging role of next-generation sequencing for laboratory diagnosis of Coxiella burnetii," has been formally accepted for publication in PLOS Neglected Tropical Diseases.

Best regards,

Shaden Kamhawi

co-Editor-in-Chief

Paul Brindley

co-Editor-in-Chief
